# Characterizing rhizome bud dormancy in *Polygonatum kingianum*: Development of novel chill models and determination of dormancy release mechanisms by weighted correlation network analysis

**Yue Wang**[1], **Donovan C. Bailey**[2], **Shikai Yin**[1], **Xuehui Dong**[1]*

**1** Department of Agronomy and Biotechnology, China Agricultural University, Beijing, China, **2** Department of Biology, New Mexico State University, Las Cruces, New Mexico State, United States of America

* xuehuidong@cau.edu.cn

**Data Availability Statement:** All original SRA files are available from the NCBI database (accession

## Abstract

This study was conducted to explore specific chill models and the mechanisms underlying rhizome bud dormancy break in *Polygonatum kingianum*. Rhizome buds were subjected to various chilling temperatures for different duration and then transferred to warm conditions for germination and subsequent evaluation of their response to temperature and chilling requirements. A $CU_{kingianum}$ model was constructed to describe the contribution of low temperature to the chill unit, and it was suggested that 2.97˚C was the optimum temperature and that 11.54˚C was the upper limit for bud release. The $CAS_{kingianum}$ model showed the relationship between chilling accumulation and sprouting percentage; therefore, rhizome bud development could be predicted through the model. Weighted correlation network analysis (WGCNA) of transcriptomic data of endo-, eco- and nondormant rhizome buds generated 33 gene modules, 6 of which were significantly related to bud sprouting percentage. In addition, 7 significantly matched transcription factors (TFs) were identified from the promoters of 17 "real" hub genes, and *DAG2* was the best matched TF that bound to AAAG element to regulate gene expression. The current study is valuable for developing a highly efficient strategy for seedling cultivation and provides strong candidates for key genes related to rhizome bud dormancy in *P. kingianum*.

## Introduction

*Polygonatum kingianum* Collett & Hemsl is an important traditional Chinese medicine [1]. However, several factors can inhibit the production of *P. kingianum* following seed germination. One of these issues derives from the stages of early rhizome development, wherein the young germinated seed quickly develops a small rhizome that enters a prolonged state of bud dormancy, requiring a sustained period of cold for the initiation of additional growth and development [2, 3]. Such cold treatment requirements have been reported for many aspects of

number SRP149787). Other relevant data are within the manuscript and its Supporting files.

**Funding:** The author(s) received no specific funding for this work.

**Competing interests:** The authors have declared that no competing interests exist.

plant growth and development, including seed dormancy, bud break, rhizome dormancy and corm dormancy [4, 5].

Several chill models have been developed to help estimate the chilling requirements for plant growth and development; however, most of these models are based on the growth of fruit tree species. The "chilling hours" model, proposed by Weinberger [6], was the first such model. It used the number of hours of exposure between 0 and 7.2˚C during the winter season, but the chilling effectiveness varied with the application of different chilling temperatures, so scholars put forward the concept of the chilling unit. The chilling unit accounts for the impact of different temperatures across time, helping researchers focus on specific practicable chilling requirements for dormancy breaks. Four classical chilling models have been proposed: the Utah model [7], low chilling model [8], North Carolina model [9] and positive Utah model [10]. Rhie adopted the cumulative CU model for evaluating the chilling requirements of *Paeonia lactiflora*, converting temperature and chill unit into a formula [11]. Fishman proposed a dynamic model, based on the counteracting effect of high temperature and low temperature treatments [12], but this model has rarely been used. Although many chilling models have been proposed, Sunley *et al.* [13] pointed out that no general models can predict bud development with sufficient precision, so it was determined that chilling models should be developed for each individual species.

Weighted correlation analysis (WGCNA) uses Pearson's correlation coefficients to measure the relationship between genes and then construct gene modules; genes in the same module have similar patterns. Hub genes in each module have important significance associated with phenotype [14]. In medical research, key target genes for the treatment of glioblastoma were identified through WGCNA [15]; additionally, target genes for treatment were identified in studies of Alzheimer's disease and osteoporosis [16]. In *Arabidopsis*, gene modules involved in seed germination were identified through WGCNA, and their underlying mechanism was further studied [17]. In apples, the gene modules associated with malic acid production were identified by WGCNA [18]. Therefore, WGCNA is very useful for mining key genes.

Transcription factors (TFs) regulate gene expression by binding to promoters of target genes and act as transcriptional activators or repressors during various stages of plant development [19]. Ren reported that the expression of *MdCIbHLH1* was up-regulated but then was gradually down-regulated during dormancy release in apple buds [20]. Yang [21] revealed that *RVE1* could promote *RGL2* stability; furthermore, *RGL2* could enhance the transcriptional activity of *RVE1* to control seed dormancy and germination in *Arabidopsis*. *DAG1* was reported to be a transcriptional repressor [22] and was shown to bind to *AtGA3ox1* to regulate seed germination [23]. Overall, TFs obviously play an important role in regulating seed dormancy. Currently, TFs related to dormancy have not been reported in *P. kingianum*. In the current study, we identified possible TFs that regulate the expression of hub genes to lay a foundation for future studies on the mechanism of dormancy release.

The process of *P. kingianum* seed emergence is as follows (S1 Fig): after the seeds were cultivated for 20 days in the dark at 25˚C, the hypocotyl broke through the seed coat (S1B Fig), after which it gradually developed to a rhizome and grew to approximately 2.5–3 cm in length (S1D Fig). A bud forms at the end of the rhizome near the seed (S1E Fig) and enters a dormancy state when it extends to approximately 3–6 mm in length (S1F Fig). However, bud dormancy can be classified as paradormancy, endodormancy or ecodormancy. Paradormancy is controlled by physiological factors that do not include the bud itself (such as apical dominance) and unless these factors are removed, the dormant bud will not resume growth [24]. Endodormancy is controlled by the physiological factors of the bud itself, and the buds need to sense certain environmental conditions, such as light and low temperature to resume growth [25]. Ecodormancy is induced by inappropriate environmental conditions such as cold or

drought stress, and once such factors are removed, bud growth resumes [25]. In our previous study [26], we identified three dormant stages induced by cold temperature for *P. kingianum* rhizome buds. After the rhizome buds were subjected to chilling treatment in the dark at 4˚C for 0 and 15 days (S1F Fig), they entered an endodormancy state, and after they were subjected to a chilling treatment for 90 days, the rhizome buds entered an ecodormancy state. Nondormant buds were obtained after the ecodormant buds were cultivated for 15 days at 25˚C under a 14:10 h light: dark period. These three different types of dormant buds were used for subsequent analysis in this research. Although transcriptomic data of endo-, eco- and nondormant buds have been reported by our team, the information obtained from the transcriptomic data is often fragmented, and gene connections between different treatments might be ignored.

In this study, we tried to develop a specific chilling model that could accurately predict rhizome bud development and speculated about the regulatory mechanism underlying how key gene modules affect bud dormancy release in *P. kingianum*. This is the first time that WGCNA and TF analysis were applied to study rhizome bud dormancy based on the whole transcriptome of *P. kingianum*.

## Materials and methods

### Plant materials

*P. knigianum* fruits (S1A Fig) were collected from Hongpo village, Deqin County, Yunnan Province, China (27˚48′46.03″ N, 99˚47′55.12″ E) and identified by Professor Xuehui Dong, College of Agronomy and Biotechnology, China Agricultural University. The seeds were obtained after the pericarp was removed and allowed to dry naturally indoors. The seeds displaying poor maturity were removed, after which plump, healthy, uniform seeds were selected and incubated at 4˚C in the dark. Wet sand (disinfected river sand plus distilled water) with a water content of 20% was prepared; afterwards, the seeds and wet sand were mixed together at a ratio of 1:7, and the mixture was put into the germination boxes [26]. The seeds were cultivated in a dark growth chamber at 25˚C for 50 days, during which time they were watered every two days. All rhizome buds that were 3–6 mm in length (S1F Fig), which had entered the dormant state, were chosen for subsequent use.

### Ethical approval

The field study was supported by Yunnan Drug Regulatory Administration and after obtaining the permission of the land owners, we purchased *P. kingianum* fruits from them, in addition, no other specific permissions were required for the collecting, since both the *P. kingianum* and the land were privately owned. The field studies did not involve endangered or protected species.

### Chill unit confirmation

Chill unit can be used to describe the efficiency of each cold temperature on releasing bud dormancy and it was studied as follows. Twenty rhizome buds were seeded at a uniform depth into trays filled with substrate (peat soil: vermiculite = 3:1). For each treatment temperature (-2, 0, 2, 4, 6, 8, 10, 12, and 14˚C), each tray was subjected to 5, 11, 15, 18, 22, 27, 32, 36, 52, 61, 71 and 81 days in the dark and watered every two days. Each treatment was repeated three times. After the treatment, the rhizome buds were transferred to 25˚C conditions with a humidity of 60%, a light intensity of 15000 Lx and a 14:10 h light:dark period for sprouting. The sprouting percentage in each treatment group was recorded daily for 70 days. Afterwards, chill unit was calculated for each temperature on the basis of the average slope of all the chilling-response curves (Fig 1) under each chilling temperature. We used the chill unit for each

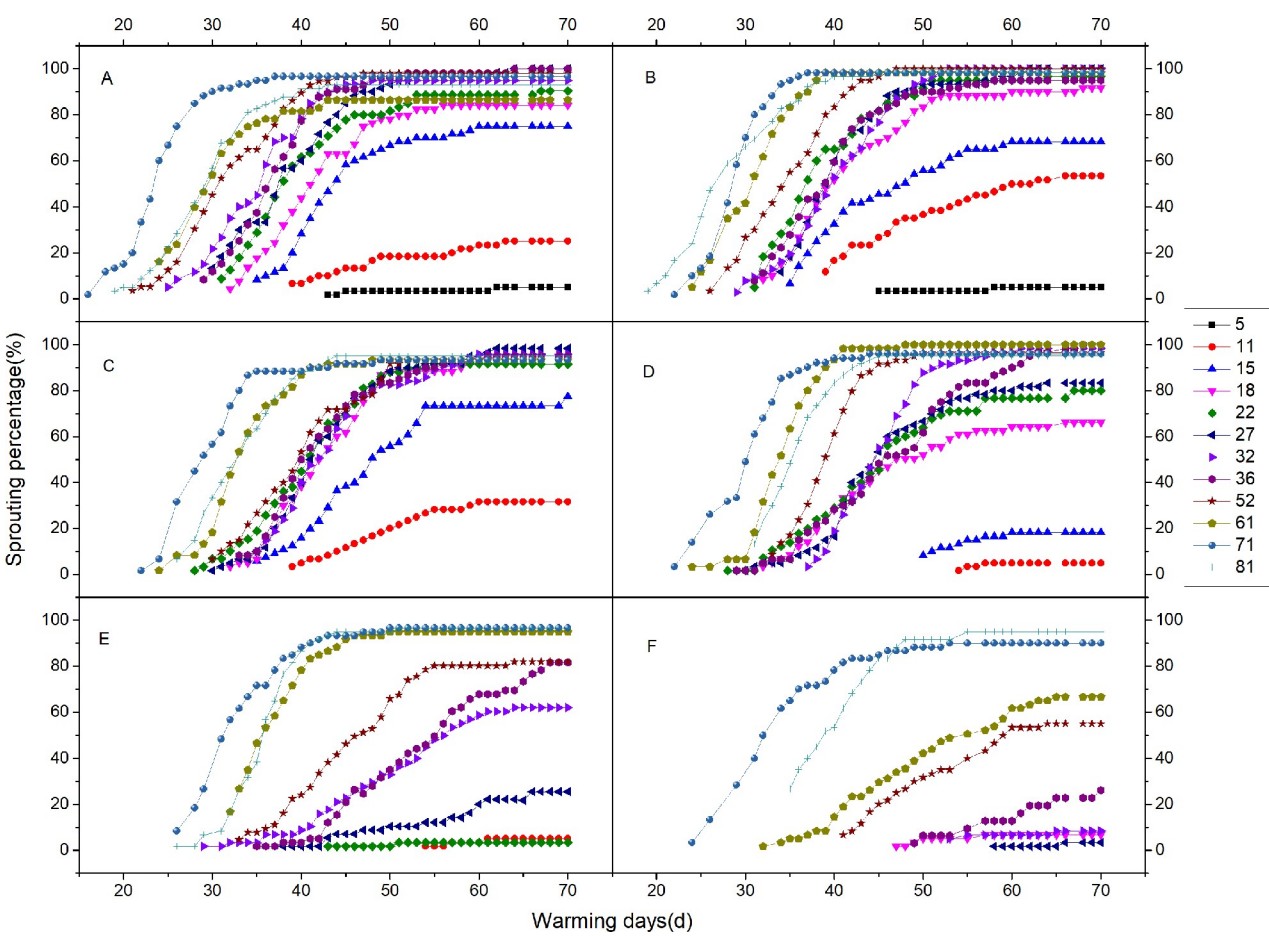

**Fig 1. Bud development under warm conditions after different chilling treatments.** (A) 0˚C. (B) 2˚C. (C) 4˚C. (D) 6˚C. (E) 8˚C. (F) 10˚C. The different line symbols represent different chilling days at each temperature.

temperature relative to the chill unit for 0˚C, after which the equation for the relationship between temperature and chilling unit was constructed. The resulting model was named $CU_{kingianum}$ (http://dx.doi.org/10.17504/protocols.io.bduhi6t6)

## Relationship between chilling accumulation and sprouting percentage

We define the chilling accumulation for *P. kingianum* rhizome buds within an hour at a certain temperature as being equal to the chill unit for the temperature obtained above. Therefore, after determining the chill unit, we substituted the chill unit into all of the above chilling treatments to obtain the chilling accumulation under each treatment. Afterwards, the function between chilling accumulation and the average sprouting was constructed. This model is referred to as the $CAS_{kingianum}$ model. Using this model, we can develop an accurate chilling strategy for the growth of *P. kingianum* rhizome bud.

## Verification of the chilling models

To verify the utility of the chilling models under more environmentally realistic variability, forty rhizome buds that were seeded at a uniform depth in trays filled with substrate (peat soil: vermiculite = 3:1) were placed in a controlled environment in which the temperature increased from 0 to 10˚C at the increments of 1˚C/day, and the increased temperature was maintained

for 88 days. One hundred and twenty rhizome buds were transferred to 25˚C conditions with a humidity of 60%, a light intensity of 15000 Lx and a 14:10 h light:dark period for sprouting every 11 days, and the sprouting percentage was checked daily for 70 days. The observed values were compared with the predicted sprouting percentages predicted by the above mentioned CAS$_{kingianu}$ model. Each treatment was replicated three times.

## Data collection for WGCNA

The transcriptomic data for endo-, eco- and nondormant rhizome buds that were previously reported by our team (National Center for Biotechnology Information (NCBI) Sequence Read Archive (SRA) accession No. SRP149787) [26] were used for WGCNA. The 4726 shared differentially expressed genes (DEGs) (fold change ≥ 2, false discovery rate (FDR) ≤ 0.05) (S1 Table) shared between the endo- vs eco-, endo- vs non- and eco- vs nondormant rhizome buds were chosen for subsequent coexpression network construction.

## WGCNA

The "WGCNA" package in R was applied to construct a scale-free coexpression network for the 4726 shared DEGs [27]. The soft threshold, β, was calculated by the pickSoftThreshold function in the WGCNA package to make the coexpression network conform to the scale-free topology. The function block wise Module was used to construct modules via the parameter deep Split 4. To identify the modules that are significantly associated with trait, that is, sprouting percentage in the current research (**S2 Table**), the correlation coefficients between the module eigengene (ME) and sprouting percentage were calculated. The correlation coefficients ranged from -1 to 1 and reflected the association between modules and traits, and the modules with correlation coefficients greater than 0.9 were defined as significant modules. The associations of an individual gene and sprouting percentage were quantified by the gene significance (GS), which was defined as the correlation between the gene and the trait. We also defined the module membership (MM) as the correlation between the module and the gene expression profile for each module. Genes in the significant modules were imported into Cytoscape_v3.6.0 for visualization of their gene coexpression network [28]. The degree reflects the number of genes that interact with a given gene, and K-mean analysis was used to determine the clustering of degree. The genes with degrees clustered into the first cluster, and genes whose GS was ≥ 0.95 and whose MM was ≥ 0.98 were identified as real hub genes. (http://dx. doi.org/10.17504/protocols.io.bdwji7cn)

## Enrichment network analysis

The enrichment network analysis for the significant modules was performed with KEGG pathways and GO database using Metascape [29](http://metascape.org/). Terms with a *P*-value < 0.01, a minimum count of 3, and an enrichment factor > 1.5 were collected, and terms with a similarity > 0.3 were grouped into the same cluster. Circos was used to display functional relationships between genes. Identical genes from different modules were linked, as were genes that were associated with the same enriched ontology term.

## Motif-based sequence alignment analysis

The sequences of the real hub genes were extracted and annotated via BLASTx (NCBI; http://www.ncbi.nlm.nih.gov/) to *Arabidopsis* and then subsequently filtered according to the criterion of E-Value < 1e-5; afterward, the promoters (3000 bp upstream from the transcript start site ATG) of the annotated hub genes in *Arabidopsis* were extracted and inputted into MEME

([http://meme-suite.org/tools/meme](http://meme-suite.org/tools/meme)) [30, 31] to perform a motif alignment. The motifs with an E-value < 0.05 were selected and subjected to the TomTom tool to identify significantly matching TFs that have functional information [32]. The Protein Data Bank (PDB) was used to retrieve the information from the matched results. ([http://dx.doi.org/10.17504/protocols.io.betkjekw](http://dx.doi.org/10.17504/protocols.io.betkjekw))

### Quantitative real-time PCR (qRT-PCR) analysis

Total RNA of the endo-, eco-, nondormant and differentially chilled (15, 30, 45, 60, 75 or 105 days) rhizome buds was extracted with TRIzol reagent (Invitrogen, Carlsbad, CA, USA) following the manufacturer's instructions. First-strand cDNA was synthesized from 2 μg of total RNA using oligo(dT) as template primers in conjunction with a SuperScript™ IV kit (Thermo Fisher Scientific) and then diluted to 100 μl for qRT-PCR amplification. The gene for *UBQ7* (forward primer: 5′-ACCCCTTGTAATACCAGTGAC-3′, reverse primer: 5′-AATAGCAG GTCGGTTTCC-3′0) served as the internal gene [26]. The target gene for *DAG2* (forward primer: 5′-TTGAACTCATAAGATCCGGAG-3′, reverse primer: 5′-TTGCTTGGCAAGTA ATGCAC-3′) was analyzed. The primers were designed via Primer 5.0 and Oligo 7 on the basis of the transcriptomic data of *P. kingianum* rhizome buds. qRT-PCR was performed on an Applied Biosystems™ 7500 Fast system. The PCR mixture comprised 10 μl of SYBR Green Master Mix (Thermo Fisher Scientific), forward and reverse primers (0.5 μM each), and 1 μl of cDNA template, and the total volume was 20 μl. The cycling parameters were as follows: denaturation at 94°C for 30 s followed by 45 cycles at 94°C for 12 s, 60°C for 30 s and 72°C for 40 s. Three biological replicates were included for all reactions. The relative expression was measured via the $2^{-\Delta\Delta Ct}$ method [33].

## Results

### Rhizome bud development after chilling treatments

Treating rhizome buds with 12°C and above had no effect on breaking rhizome bud dormancy for three months after treatment. The -2°C treatment resulted in damage to the buds; thus, we did not consider these treatments further. It is apparent from Fig 1 that the initial emergence time becomes delayed and that the slope of the curve decreases with increasing temperature. The earliest bud break (regardless of whether or not the final sprouting percentage reached 60% within 70 days was taken as a sign of dormancy release) after the 0, 2, and 4°C treatments occurred after 15 days (Fig 1A–1C). The 6, 8, and 10°C treatments resulted in the first signs of dormancy break after 18, 32, and 61 days, respectively (Fig 1D–1F). With respect to the 0°C treatment, the cumulative sprouting percentage reached stabilized after 37–60 days, with final levels varying from 75% of those of the 15 days chilling treatment to 98.04% of those of the 52 days chilling treatment. Furthermore, the cumulative sprouting percentage increased with increasing chilling duration, and all the 0°C-chilling treatments presented rather similar trends, although with different levels at each time point. Interestingly, the 0°C 71 days chilling treatment had the greatest sprouting percentage at most time points (Fig 1A). This phenomenon also occurred at the other chilling temperatures (Fig 1B–1F). According to the emergence uniformity and the final emergence rate, the best chilling treatment was 71 days at 2°C.

### Comparison of rhizome bud development among different chilling temperatures

To determine the response of rhizome bud development to the various chilling temperatures and duration, we showed the sprouting percentage after 37, 40, 44 and 50 days of warm

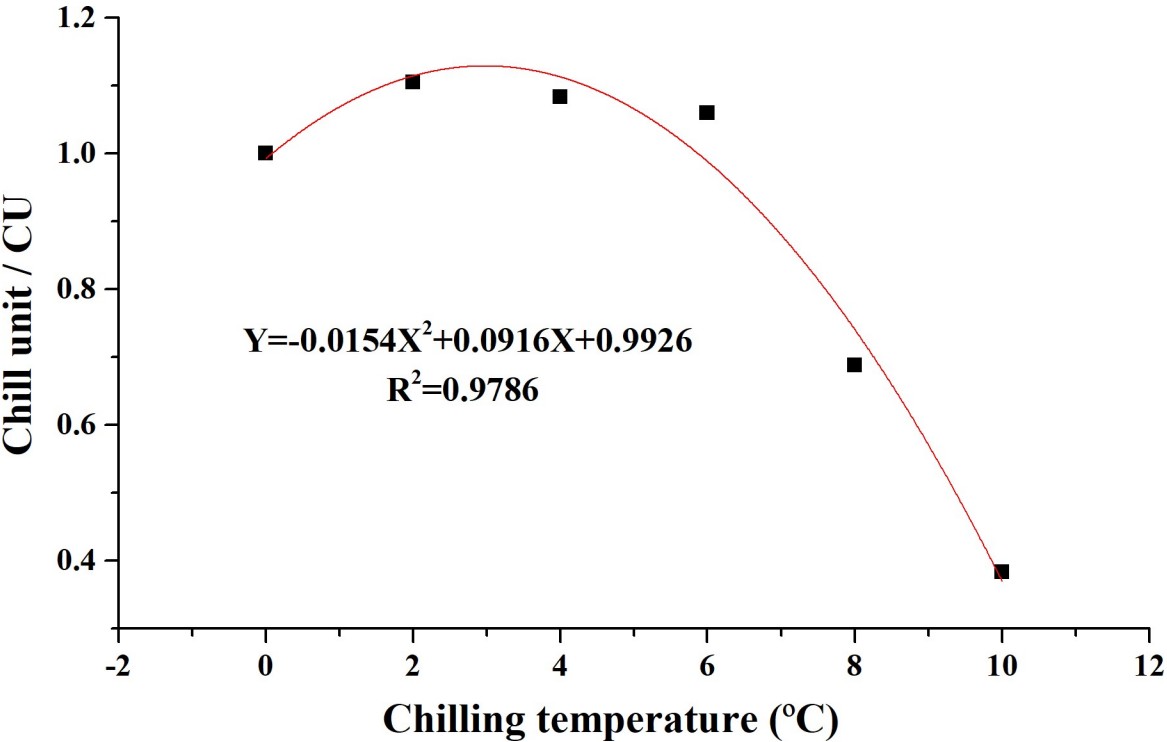

**Fig 2. The CU$_{\text{kingianum}}$ model describing the contribution of low temperature to rhizome bud dormancy break in *P. kingianum*.** The x-axis (Chilling temperature) is the cold temperature used to release bud dormancy. The y-axis (Chill unit), whose values are calculated from the average slope of all the chilling-response curves under each chilling temperature, represents the efficiency of a particular cold temperature on the release of bud dormancy, and the unit (CU) is assigned. The greater the chill unit value is, the greater the promotion effect of the cold temperature on the bud dormancy release.

treatment (S2 Fig). The results showed that the differences among the chilling temperatures decreased with increasing chilling duration. Furthermore, with respect to the 2–10˚C cold treatments, there was a general inverse relationship between the cold treatment temperature and the percentage of bud break. However, the 0˚C treatment sometimes resulted in a lower percentage of bud break than did the 2˚C treatment. Interestingly, the sprouting percentage showed slightly decreased after the rhizome buds were treated for 71 days at all temperatures used.

### Derivation and verification of chilling models for rhizome bud development

It is apparent that the CU$_{\text{kingianum}}$ model (Fig 2) best fits the function Y = -0.0154X$^2$ + 0.0916X + 0.9926 (R$^2$ = 0.9786), where X (0 $\leq$ X $\leq$ 10) represents the chilling temperature, and Y represents the chilling unit for that temperature. The results suggested that the optimum chilling temperature for rhizome bud dormancy release is 2.97˚C, with the greatest chilling unit of 1.1288 CU. The CAS$_{\text{kingianum}}$ model (Fig 3) describes the relationship between chilling accumulation and sprouting percentage on the 37$^{\text{th}}$, 40$^{\text{th}}$, 44$^{\text{th}}$ and 50$^{\text{th}}$ warming day. The results indicated that the optimum chilling requirement varied among the sprouting stages, with optimum chilling requirements of 4664 Ca at 37 warming days, 2548 Ca at 40 warming days, 1764 Ca at 44 warming days and 1438 Ca at 50 warming days.

In the verified experiments, the bud sprouting percentage increased with chilling duration, but we found that the seedlings exposed to cold temperature for 66 days were slightly stronger

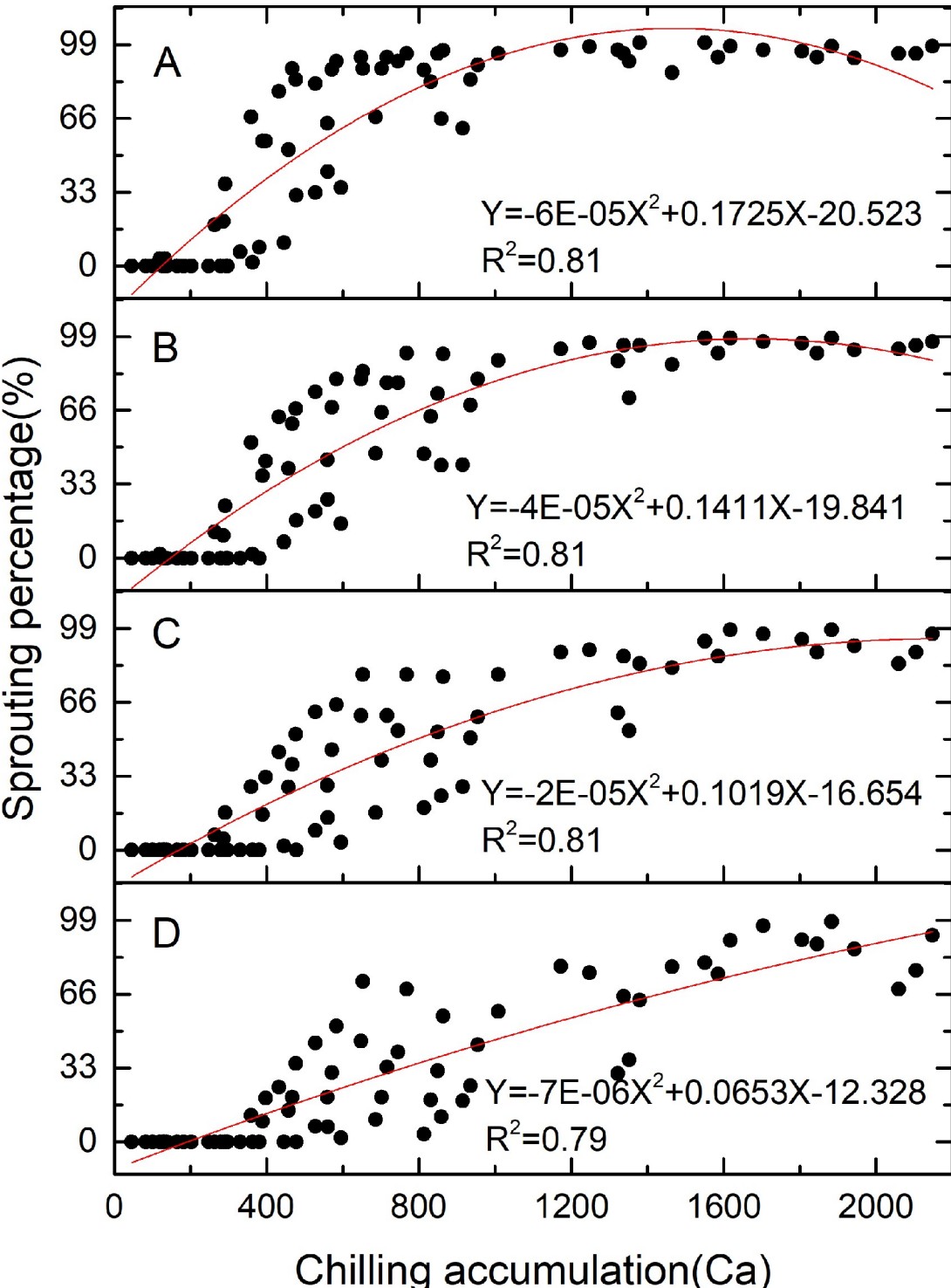

**Fig 3. CAS$_{kingianum}$ models describing the relationship between chilling accumulation and sprouting percentage at four bud stages.** (A) The 37th warming day. (B) The 40th warming day. (C) The 44th warming day. (D) The 50th warming day.

than those exposed for 77 and 88 days (Fig 4). The CAS$_{kingianum}$ modeled sprouting percentage was checked against the data of the verified experiments (Fig 5). The CAS$_{kingianum}$ values

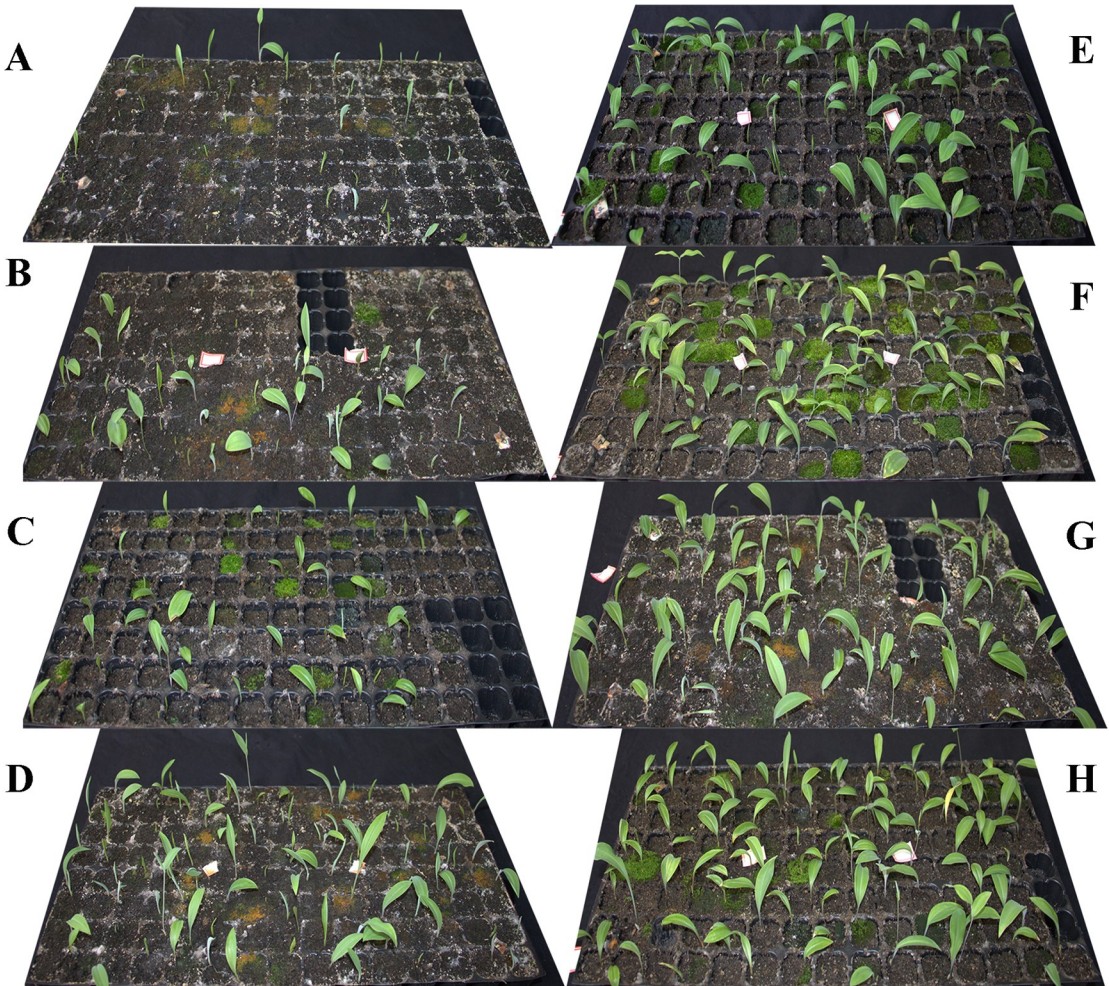

**Fig 4. Rhizome buds on the 40<sup>th</sup> warming day after different chilling treatments in the verification experiment.** (A) Eleven chilling days. (B) Twenty-two chilling days. (C) Thirty-three chilling days. (D) Forty-four chilling days. (E) Fifty-five chilling days. (F) Sixty-six chilling days. (G) Seventy-seven chilling days. (H) Eighty-eight chilling days.

approximated the observed values. Interestingly, the observed values were slightly greater than the predicted values at all four bud stages (**Fig 5**), but a strong correlation was detected between the observed values and predicted values ($R^2 > 0.95$). The fitted line approximates the equation $Y = X$ (**S3 Fig**).

## Thirty-three coexpression modules were produced via WGCNA

Nine samples were clustered, and a cluster diagram (**S4 Fig**) was constructed to determine the correlations among the samples and to determine the correlations between the genes and their corresponding traits. The results showed that there were no outlying samples, and the samples were strongly correlated with the sprouting percentage. Herein, the power of $\beta = 26$ was chosen to construct a scale-free network; under this parameter, the $R^2$ value for the topology was 0.87, and the mean connectivity was 495 (**S5 Fig**). Thirty-three modules were identified after the analysis and represented by different colors (**Fig 6**). The largest turquoise module contained 409 genes, and the smallest violet module contained only 42 genes. The details for each module are shown in S1 Table.

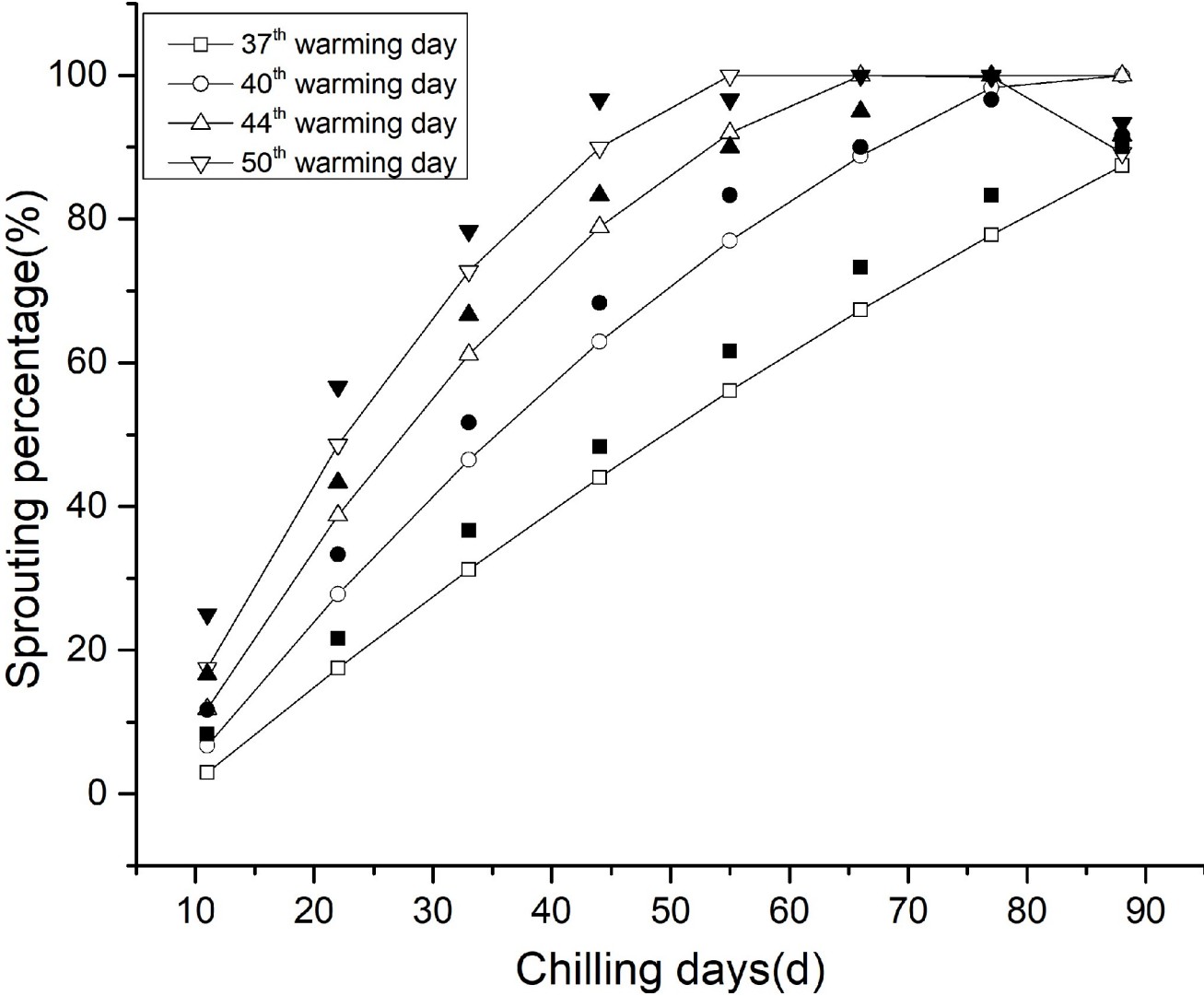

**Fig 5. Validation of the CAS$_{kingianum}$ model on the 37th, 40th, 44th and 50th warming day.** The sprouting percentage we obtained from the verification experiment and the developed CAS$_{kingianum}$ models are represented by solid and hollow shapes, respectively. The same shape represents the same warming day.

### Identification of key modules for the dormancy transition

The relevance between module and trait was further evaluated. The results showed that the dark red, royal blue, sky blue, violet, tan and white modules, with module-trait associations > 0.9, were significant modules; these modules were used in subsequent analyses (**Fig 7**). Genes in the tan module were enriched mainly in terms involving the cell wall macro-molecule metabolic process, detoxification, the purine-containing compound biosynthetic process, glycolysis, tropism and the phenylpropanoid biosynthetic process; genes in the royal blue model were enriched mainly in terms involving the phenylpropanoid biosynthetic process, response to wounding and hypoxia and the cellular ketone metabolic process; genes in the violet module were enriched mainly in terms involving anion transport; genes in the white module were enriched mainly in terms involving gametophyte development; genes in the dark red module were enriched mainly in terms involving plastid organization; and genes in the sky

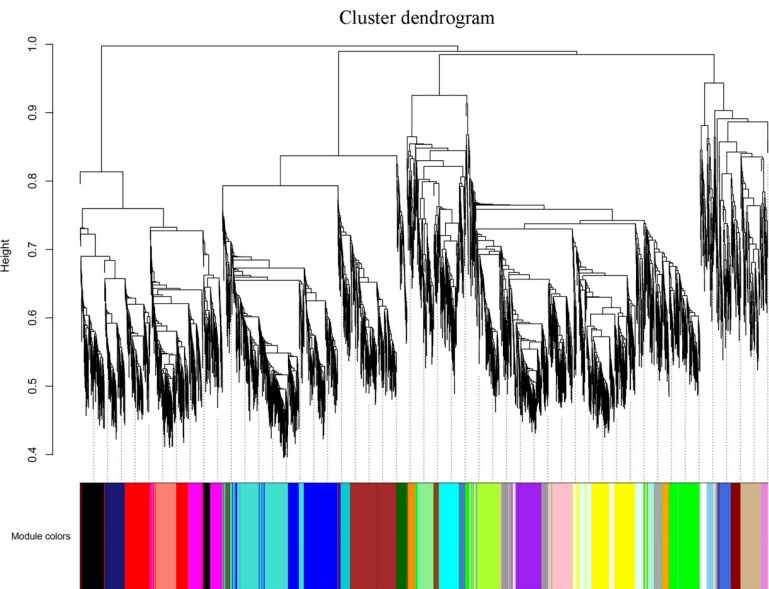

**Fig 6. Cluster dendrogram of 4726 DEGs together with their assigned module colors.** The y-axis (height) represents the similarity coefficients between genes.

blue module were enriched mainly in terms involving anatomical structure maturation (**Fig 8A** and **S3 Table**). We identified a shared term related to the phenylpropanoid biosynthetic process between the tan module and royal blue module. These terms were divided into 13 clusters, and there were some shared genes among terms in each cluster, expressed as the connection of two nodes in the enrichment network (**Fig 8B**) and indicating the relatedness of two separate processes during dormancy release of the *P. kingianum* rhizome bud. In addition, the 6 significant modules shared several terms related to the hormone biosynthetic process, abscisic acid metabolic process, stomatal movement and the homeostatic process and shared several genes, indicating similar developmental patterns for different modules (**Fig 8C**).

## Identification of real hub genes and TFs involved in dormancy transition

The genes with edge weights > 0.4 in the tan module, > 0.3 in the dark red module, > 0.25 in the violet module, > 0.2 in the white module, > 0.25 in the sky blue module and > 0.35 in the royal blue module were imported into Cytoscape_v3.6.0 to construct a gene coexpression network (**Fig 9A**)(more details are shown in S2 Table). The genes with a GS > 0.95 and an MM > 0.98 in the tan module are shown in Fig 9B. When the above results are combined, a total of 37 hub genes were identified (S6 Fig), of which the expression of 17 genes was reduced, while that of 20 genes increased during the transition of dormancy (S6 Fig). The 17 genes

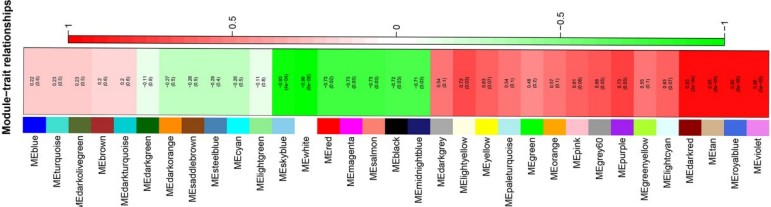

**Fig 7. Diagram of associations between gene modules and sprouting percentage.**

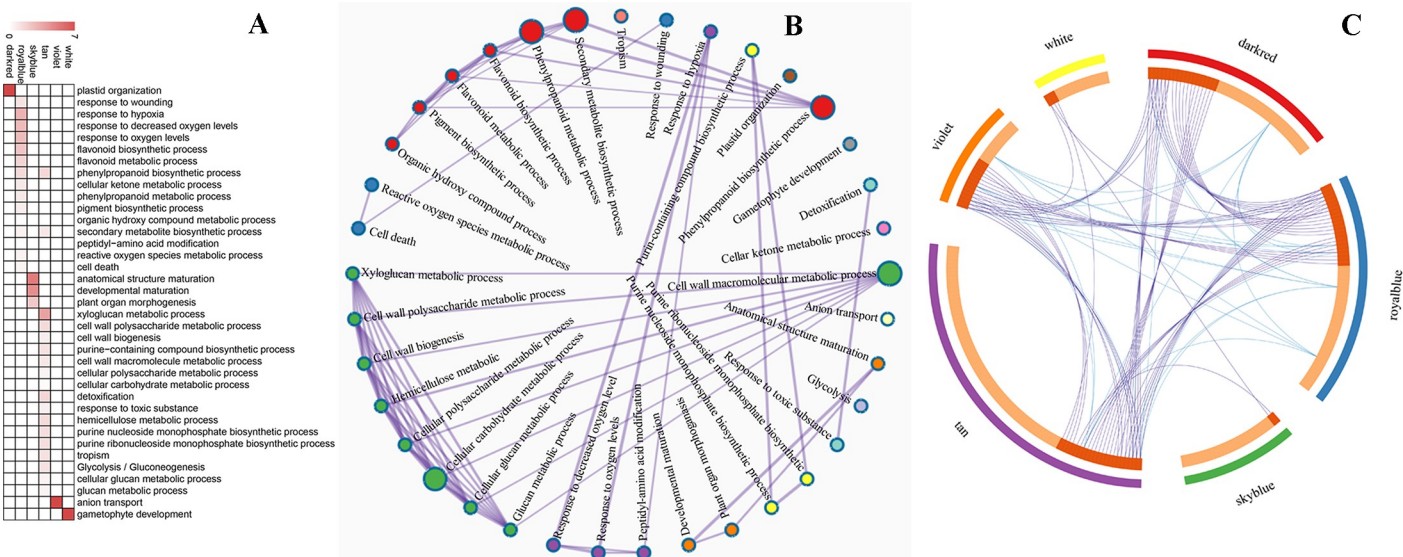

**Fig 8. Metascape functional categories and enrichment networks of 6 significant modules.** (A) Enriched terms in each significant module, and the darker the color is, the more significantly enriched the term. (B) Network of enriched terms colored by cluster ID; each node represents a term. The node size is proportional to the number of genes belonging to the term, and the node color represents the identity of the cluster. The terms with a similarity > 0.3 are connected by edges. (C) Overlap between gene lists. The purple curves are links between identical genes, and the blue curves are links between genes that belong to the same enriched ontology term. The genes on multiple lists are colored in dark orange, and genes unique to a list are shown in light orange.

whose expression is down-regulated are involved in the regulation of meristem growth, development and cell death, lysine decarboxylase, the regulation of transcription and the promotion of floral meristem fate, and the 20 genes whose expression is up-regulated are involved in oxidation-reduction, naringenin-chalcone synthase activity, response to cytokinin stimuli, and protein amino acid phosphorylation (S2 Table).

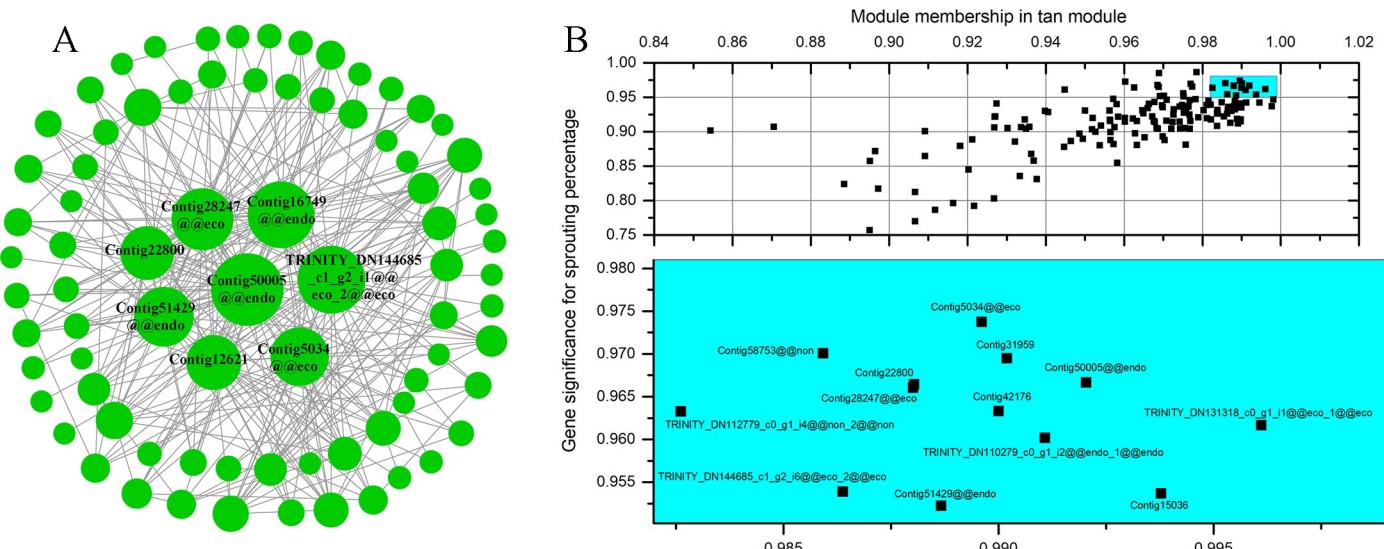

**Fig 9. Real hub genes in the module.** (A) Each node represents a gene. Genes that interact with each other are connected by lines. The node size is proportional to the number of genes that interact with that node. (B) GS and MM information of hub genes in the tan module. The genes in the blue area are the real hub genes.

A motif-based sequence alignment of 17 annotated hub genes was conducted to investigate the possible regulatory mechanism of the hub genes. Three significantly matched motifs (E-value < 0.05) with 7 matched TFs were identified (S7 Fig and S4 Table): *DAG2* (AAAG), *Dof5.7* (AAAG), *bZIP60_2* (ACGTCA), *MYB111_2* (CACC), *MYB55_2* (CACC), *MYB46_2* (CACC), and *REM1* (TGTAG). Interestingly, the best matched TF was determined to be *DAG2*, which is reportedly involved in the cold response and the control of seed germination [34]. Together, these results indicated that *DAG2* may bind to promoters of the hub genes to regulate bud break in *P. kingianum*.

## Discussion

### Confirmation of chill models of rhizome bud dormancy release in *P. kingianum*

Prior research on blackcurrant cultivar has revealed a positive contribution of chilling treatment below 0°C to bud dormancy release [13, 35]. Sunghee and Neilson [36] specified -2 to 5.5°C as the temperature range that was most effective for bud release in *Malus*, and 0–7°C was reported to be the best temperature range for rabbiteye blueberry [8]. For this study, we set the lowest temperature treatment as -2°C, but such a cold temperature caused frost damage to the rhizome buds of *P. kingianum*; thus, it was clear that this species has its own unique chilling requirements. The optimal cold temperature for bud dormancy release in *P. kingianum* was 2.97°C (Fig 2), which was similar to that found in previous studies reporting optimal chilling temperatures of 2°C and 3.2°C for '*Jonathan*' apple and sweet cherry, respectively [37, 38].

Research on the effective upper limit for the chilling temperature via the Utah model considered the maximum temperature that can contribute to break bud dormancy to be 12.4°C in peach [7] and 16°C in sweet cherry [13]. Our $CU_{kingianum}$ model indicated that the chill unit value is 0 when the chilling temperature reaches 11.54°C (Fig 2), and this was confirmed by the chilling treatments at 12 and 14°C, which had no influence on the sprouting percentage compared with those of the control treatment.

Another important finding was that excessive exposure to low temperature could weaken rhizome bud development. This effect is reflected by the fact that the sprouting percentage under the 81 days chilling treatment being lower than that under the 71 days chilling treatment at all temperatures (S2 Fig). Additionally, the seedlings exposed to chilling treatment for 66 days were slightly stronger than those exposed for 77 and 88 days (Fig 4). It is also possible that excessive chilling treatment delays the initial emergence days. This possibility explains the supra-optimal chilling effect that was also mentioned by Jones *et al.* [39]; thus, chilling duration is an important consideration in cultivation. Moreover, the observation that the 0°C treatment sometimes resulted in a lower sprouting percentage than did the 2°C treatment (S2 Fig) reveals that the lowest temperature used in this treatment may not be the best at releasing rhizome bud dormancy.

Although the chill unit for 4°C (1.083 CUs) is greater than those for 6°C (1.059 CUs), we found that the rhizome buds under 6°C sometimes had a greater sprouting percentage than did those under 4°C after cold treatment for 52, 61, 71 and 81 days (S2 Fig). These findings could be related to the satisfaction of chilling requirements. Similarly, Campoy [40] reported that the most effective temperature varied with chilling accumulation. In addition, Lavarenne *et al.* [41] also suggested that the temperature effect is related to the physiological state of the plant.

Several chilling unit models have been reported, and most of them assumed a positive correlation between temperature and chill units, but these models cannot explain why 2°C was

more efficient than 0˚C at several time points (S2 Fig) for *P. kingianum*. In the present study, we developed a $CU_{kingianum}$ model that was best fit by the function $Y = -0.0154X^2 + 0.0916X + 0.9926$, where 2.97˚C was defined as the optimum temperature, and no contribution was made above 11.54˚C. The chill unit was calculated as a bud break percentage at optimum temperature [38, 42], but bud break was not sufficient for reflecting the efficiency of different temperatures for *P. kingianum*. Therefore, we designed the chill unit for each temperature as the relative values of the slope for 0˚C-chilling-response curves in our model, making this the first study to use this method to characterize the chill unit.

We developed the $CAS_{kingianum}$ model to describe the relationship between chilling accumulation and bud development. According to this model, chilling accumulations of 1940, 1472, 1160 and 964 Ca are required at 37th, 40th, 44th and 50th posttreatment warming day, respectively, to achieve 90% sprouting. Although several models have been reported, they are not adaptable to the data in this study; however, the modeled values obtained from the $CAS_{kingianum}$ model were compared with the observed values (Fig 5). The results showed that the observed values were slightly greater than the predicted values, which might be a consequence of the "alternating temperature regime" effect [43], which states that alternating temperatures can have a greater effect on dormancy release than can constant temperature. Varying temperature was found to be more effective than a given average temperature [39]. However, the difference can be negated by linear fitting between the predicted and observed values in the current study (S4 Fig).

## Terms enriched in the significant modules

In the current study, 3 genes, *FBA6*, *AT1G32780* and *mtLPD1*, were enriched in glycolysis/gluconeogenesis terms. *FBA6* was involved in the conversion of β-D-fructose-1,6P2 and glyceraldehyde-3P. *mtLPD1* was involved in the oxidation of pyruvate, the product of glyceraldehyde, under aerobic conditions, while *AT1G32780* was involved in hypoxic conditions (according to Kyoto Encyclopedia of Genes and Genomes (KEGG) pathway analysis). The results indicated that glucose metabolism is important during the process of dormancy release in the *P. kingianum* rhizome buds. Interestingly, terms related to flavonoid biosynthetic and metabolic processes were both enriched in the six significant modules. It is known that many kinds of flavonoids have medicinal value; therefore, it is possible that flavonoids are already accumulating in the early stages of the *P. kingianum* rhizomes. By performing an enrichment analysis performed by combining the six significant modules (S5 Table), we identified several unique terms compared with those in **Fig 8**. These terms are related to hormone biosynthetic/metabolic processes, regulation of hormone levels, jasmonic acid biosynthetic processes and abscisic acid metabolic processes; therefore, these results again confirmed our previous findings: the hormone biosynthetic/metabolic processes played an important role in controlling dormancy release in the *P. kingianum* rhizome bud.

## TFs that were identified to bind hub genes to regulate gene expression

A total of 4726 shared DEGs among endo- vs eco-, endo- vs non- and eco- vs nondormant rhizome buds were chosen from approximately 203772 contigs that were previously reported by our research team ((NCBI SRA accession No. SRP149787) [26]. WGCNA was successfully implemented for the 4726 DEGs, resulting in 33 modules and 17 annotated hub genes. The hub genes were then subjected to motif alignment analysis. Three significant motifs were matched to 7 TFs representing 4 families. *MYBs* recognize CACC-containing motifs. Dof domain-containing proteins such as *DAG2* and *dof5.7* recognize an AAAG conserved sequence. *bZIP60_2* contains an ACGTCA-like domain and *REM1* recognizes TGTAG

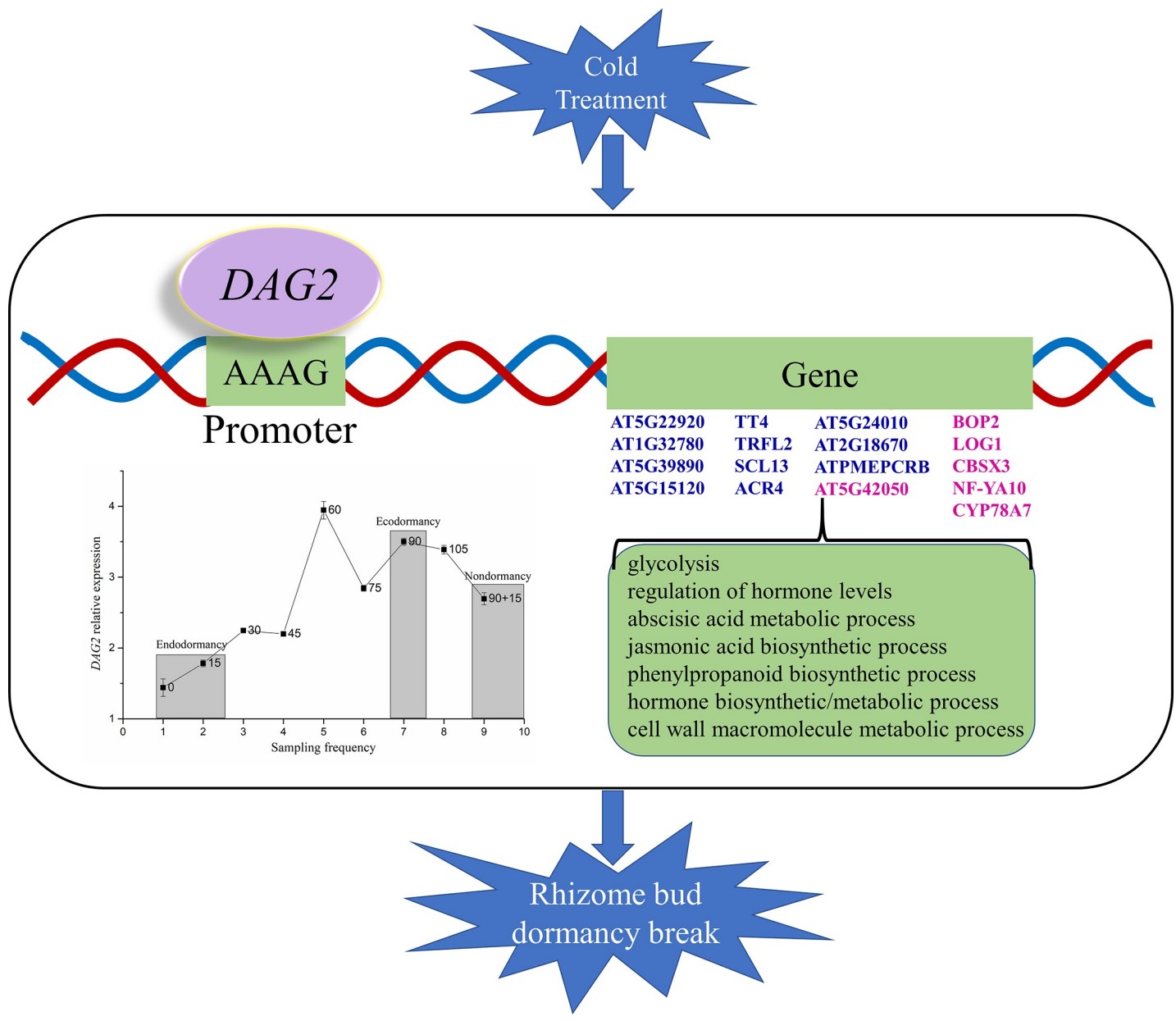

**Fig 10. Proposed model for dormancy release of *P. kingianum* rhizome buds induced by low temperature.** *DAG2* expression during the chilling process is shown on the left, and endo-, eco- and nondormant buds are shaded. The data labels represent chilling days at 4˚C, and "90+15" means 90 days of chilling followed by transfer to 25˚C for 15 days. The expression of the genes marked in blue is up-regulated, and the expression of the genes marked in red is down-regulated during rhizome bud dormancy break.

elements [19]. The best matching motif matched the *DAG2* significantly. Interestingly, *DAG2* was reported to be involved in the control of seed germination. Silvia [44] reported that *DAG2* acts as a positive regulator of light-mediated seed germination. Our research indicated that *DAG2* is associated with the regulation of hub genes during the dormancy transition induced by chilling. In line with our results, it was reported that mutant *dag2* seeds were more dependent on cold temperature than wild type seeds [34]. However, not all genes with specific elements are always target genes for their corresponding TFs [22]; therefore, our results need to

be verified with in vitro experiments. However, our analysis provides strong candidates for identifying key genes in future studies.

In the current study, 17 annotated hub genes were bound by *DAG2*, and the expression of 12 of these genes was up-regulated during dormancy release (**S2 Table**). Moreover, qRT-PCR also showed that the expression of *DAG2* was up-regulated after endodormancy was released (Fig 10). Therefore, we speculate that *DAG2* plays a positive transcriptional role in the rhizome bud of *P. kingianum*. In line with this speculation, a previous study also showed that *DAG2* acted as a transcriptional activator in an in vitro experiment [19]. Moreover, five of the 17 annotated hub genes were enriched in terms related to glycolysis, flavonoid biosynthetic/metabolic processes, phenylpropanoid biosynthetic/metabolic processes, peptidyl-amino acid modification and the response to hypoxia as well as oxygen levels according to the enrichment network analysis. There were cross-links between these terms (Fig 8). In addition, the expression of *DAG2* was up-regulated during the process of chilling-induced dormancy release and reached its greatest level on the 60th day of chilling treatment. In addition, *DAG2* expression in nondormant buds was greater than that in endodormant buds but lower than that in ecodormant buds. Summarizing the above results, we proposed a model (Fig 10) in which *DAG2*, under low temperature exposure, promotes the transcription of 37 "real" hub genes involved in several enriched terms by binding to the AAAG element within the promoter regions of real hub genes, promoting the release of rhizome bud dormancy in *P. kingianum*.

## Conclusions

It has been shown that excessive cold treatment has a negative impact on bud growth in *P. kingianum*. This is based on a single concept: the optimum chilling requirement. Therefore, we developed two chill models for *P. kingianum* bud growth: the $CU_{kingianum}$ model and the $CAS_{kingianum}$ model. The $CU_{kingianum}$ model showed that 2.97°C had the greatest contribution to bud dormancy release, and no contribution was made above 11.54°C. The $CAS_{kingianum}$ model can help us determine the chilling accumulation needed for bud growth so that an effective strategy for cultivation can be generated, and the modules indicated that chilling accumulations of 1940–964 Ca are required from the 37th to the 50th posttreatment warming day to achieve 90% sprouting, which can help guide the breeding of *P. kingianum*. Through WGCNA, 6 gene modules were found to be significantly related to dormancy release, and they were enriched in terms associated with glycolysis, hormone biosynthetic/metabolic processes and cell wall macromolecule metabolic processes. Thirty-seven real hub genes were identified, and *DAG2* was found to be the best matched TF that bound to the promoters of real hub genes involved in the dormancy release of *P. kingianum* rhizome buds.

## Supporting information

**S1 Fig. Seedling development of *P. kingianum*.** (A) The seed of *P. kingianum*. (B) Three days after the hypocotyl breaks through the seed coat. (C) The epicotyl begins to swell into a rhizome. (D-F) The rhizomes continue to swell, and roots (D) and buds (F) have formed. (F) The rhizome bud ceases growth and enters endodormancy. (G-I) The rhizome bud grows into a seedling after dormancy is released. (Wang *et al.* 2019).
(TIF)

**S2 Fig. Comparison of bud growth for different chilling temperatures at four bud stages.** (A) The 37th warming day. (B) The 40th warming day. (C) The 44th warming day. (D) The 50th warming day.
(TIF)

**S3 Fig. High correlation between actual and CAS$_{kingianum}$ modeled sprouting percentage in the verification experiment.** (A) The 37$^{th}$ warming day. (B) The 40$^{th}$ warming day. (C) The 44$^{th}$ warming day. (D) The 50$^{th}$ warming day. The gray line is Y = X.
(TIF)

**S4 Fig. The phylogenetic tree and information for the 9 samples.**
(TIF)

**S5 Fig. Analysis of network topology for various soft-thresholding powers.**
(TIF)

**S6 Fig. Expression pattern and module attribution for 37 "real" hub genes in 9 rhizome bud samples.** For each gene, the FPKM value normalized by the maximum value of all FPKM values is shown. The gene symbol for each gene is shown on the left. The module attribution for each gene is shown in the second left column and the corresponding module for each color is shown on the right.
(TIF)

**S7 Fig. Motif analysis for 17 annotated hub genes.** For Motif 1, the TF *DAG2* was significantly matched; for motif 2, two significantly-matched TFs, *REM1* and *Dof5.7*, were found; for motif 3, four significantly-matched TFs, *MYB55.2*, *bZIP_2*, *MYB46_2* and *MYB111_2*, were found.
(TIF)

**S1 Table. Expression pattern of 4726 shared DEGs in nine different dormant rhizome buds.** The expression is represented by FPKM value.
(XLS)

**S2 Table. WGCNA related information.** "Phenotype" describes the sprouting percentage of endo-, eco- and nondormant rhizome buds. "ModuleInformation" describes the module attribution for each gene and number of genes contained in each gene module. "GSandMM" describes GS and MM value for each gene. "GS.Sprouting_percentage" represents the associations of each gene with sprouting percentage, and the greater the absolute value of GS, the more correlated this gene was with the dormancy releasing. "MM" represents the correlation of the module eigengene and each gene expression profile, and the greater the absolute value of MM, the more the gene belongs to the module. "P.GS" and "P.MM" represent *P* value for "GS.Sprouting_percentage" and "MM" respectively. "Hub genes" describes the process of screening "real" hub genes from hub genes and the genes marked with red are the "real" hub genes. "Real hub genes" describes the module attribution and annotation for each "real" hub gene. "Expression profile of hub genes" describes the FPKM in different dormant buds and functional annotation for each "read" hub gene.
(XLSX)

**S3 Table. Information related to enrichment network analysis for the genes in each significant modules.** "Annotation" describes the module attribution and GO annotation for each gene. "Enrichment" describes the statistic information for each enriched term.
(XLSX)

**S4 Table. Motif alignment analysis for the 17 annotated hub genes.** Through the alignment for the 17 promoter sequences, a total of 3 significantly-matched motifs were found.
(DOCX)

**S5 Table. Information related to enrichment network analysis for the whole genes in 6 significant modules.** "Annotation" describes the module attribution and GO annotation for each gene. "Enrichment" describes the statistic information for each enriched term.
(XLSX)

# Acknowledgments

The work was supported by Chinese Agricultural University. The author thanks Professor Dong for providing guidance on the research idea and experimental facilities. We also thank Professor Bailey of New Mexico State University for language editing.

# Author Contributions

**Data curation:** Yue Wang.

**Formal analysis:** Yue Wang.

**Funding acquisition:** Xuehui Dong.

**Investigation:** Yue Wang, Shikai Yin.

**Project administration:** Xuehui Dong.

**Writing – original draft:** Yue Wang.

**Writing – review & editing:** Donovan C. Bailey.

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
