## [Decision Letter · Decision Letter 0]

5 Mar 2020

PONE-D-20-00745

Characterizing rhizome bud dormancy in Polygonatum kingianum: chill models and mechanisms revealed by weighted correlation network analysis.

PLOS ONE

Dear Dr. Wang,

Thank you for submitting your manuscript to PLOS ONE. After careful consideration, we feel that it has merit but does not fully meet PLOS ONE’s publication criteria as it currently stands. Therefore, we invite you to submit a revised version of the manuscript that addresses the points raised during the review process.

We would appreciate receiving your revised manuscript by Apr 19 2020 11:59PM. To enhance the reproducibility of your results, we recommend that if applicable you deposit your laboratory protocols in protocols.io, where a protocol can be assigned its own identifier (DOI) such that it can be cited independently in the future. For instructions see: http://journals.plos.org/plosone/s/submission-guidelines#loc-laboratory-protocols

We look forward to receiving your revised manuscript.

Kind regards,

Alexandre Fournier-Level, Ph.D.

Academic Editor

PLOS ONE

Journal Requirements:

Additional Editor Comments (if provided):

Dear Authors,

Thank you for submitting your article to PLoS ONE.

Your work has been reviewed by three reviewer who all find merit to your study and wish to see you work ultimately published in our journal.

However, the reviewers have pointed to a few aspects that need improvement in particular:

- Tighten the abstract and the end of the Introduction to make the aim of the study very clear (abstract and intro) and simplify/shorten the abstract to focus on the key result.

- Describe the plant material and the experimental design in greater detail using comments from Reviewers 1 and 2.

- Revise the reefernce to adhere to teh PLoS format

Please disregard comments on figure quality as the compressed version of the paper provided to the reviewer was bad but the figures you submitted are good enough.

Sincerely

Reviewers' comments:

Reviewer's Responses to Questions

**Comments to the Author**

1. Is the manuscript technically sound, and do the data support the conclusions?

Reviewer #1: Yes

Reviewer #2: Yes

Reviewer #3: Yes

2. Has the statistical analysis been performed appropriately and rigorously? 

Reviewer #1: Yes

Reviewer #2: I Don't Know

Reviewer #3: Yes

3. Have the authors made all data underlying the findings in their manuscript fully available?

Reviewer #1: Yes

Reviewer #2: Yes

Reviewer #3: Yes

4. Is the manuscript presented in an intelligible fashion and written in standard English?

Reviewer #1: Yes

Reviewer #2: Yes

Reviewer #3: No

5. Review Comments to the Author

Reviewer #1: In this study, the authors conducted a series of experiments on the rhizome buds of Polygonatum kingianum to explore specific chill models and mechanisms for rhizome bud dormancy break. A “CUkingianum” model was deduced to describe the contribution of low temperature to the chill unit, and the rhizome bud development can be predicted through the “CASkingianum” model. Besides, weighted correlation network analysis (WGCNA) was used to study transcriptomic data of endo-, eco- and nondormant rhizome buds, and a network containing 33 modules was constructed. Those results provide insights into the mechanisms for rhizome bud dormancy break in P. kingianum, which has a certain significance for production practice. The manuscript was well written in English. However, a revision of manuscript is required before it can be accepted for publication. Please see next comments.

1. Most figures are not clear, if possible, please indicate more clear photos.

2. The dormant situation in the place of origin and environmental conditions should be provided in order to understand the main factors leading to dormancy.

3. Line157-158, the best treatments should be only one.

4. Line194-195, A, B, C and D should be shown in Fig 4.

5. Line201, “are” should be deleted.

6. Line204, “Clustering dendrogram” should be consistent with that in Fig 5, the Y axis Height should give an explanation in figure legend.

7. Line231, the title does not match what follows, it should be revised.

8. Line252-253, “The results indicated that DAG2 may bind to promoters of the hub genes to regulate bud germination in P. kingianum.” is just a speculation and lack of experimental evidence. Therefore, qRT-PCR analysis of DAG2 expressed in rhizome buds of P. kingianum during chilling and germination processes should be supplemented.

9. Line427, reference 25 (Peter LH, S. Tutorial for the WGCNA package for R: UC Los Ageles; 2014 [) is not cited correctly.

10. In Supplementary Fig. 3, “VS” should be lower case letters.

In addition, the reference format should meet the requirements of PLoS One.

Reviewer #2: Polygonatum bud dormancy has been investigated based on qRT-PCR, correlation and network analysis and few models have been proposed. Manuscript largely appears to be accepted for publication after answering following queries and implementing few suggestions:

1. Grammar needs to be thoroughly checked for example line 142.

2. Line 75 does term mean GO term?

3. Under methods section, more details about growth conditions such as humidity, watering, light intensity, duration, and weather seeds were grown in soilrite and what media was provided?

4. Line 99: Was day/night temperature same?

5. How authors classified dormancy as endo or eco or no dormancy. Provide some details.

6. Line 132: How transcription start site was determined?

7. Line 136: What tissue was used for qPCR analysis?

8. Line 139 Instead use qRT-PCR or qPCR everywhere in MS

9. Format supplementary tables and label title clearly.

Reviewer #3: Manuscript (PONE-D-20-00745) “Characterizing rhizome bud dormancy in Polygonatum kingianum: chill models and mechanisms revealed by weighted correlation network analysis” by Wang et al. presents a study about rhizome dormancy in Polygonatum. This manuscript presents a new methodology applied to one Polygonatum kingianum genotype with several cold-dormancy conditions.

The interest of the work is high due to the importance of the studied process. However this manuscript presents several deficiencies which must be revised and clarified before publication in PLoSOne Journal.

The major points for the revision of the manuscript are:

Abstract should be reduced only indicating the key information.

In the Introduction, first paragraph (lines 27-47) should be separated in at least two paragraphs to improve understanding.

Objectives of the work should be better clarified eliminating the references (page 4 line 96).

Description of methodology should be improved. The Plant material used must be clarified indicating if it is possible any reference. Description of RT-PCR must be completed. It is very important the description of the assayed genes.

Figure S2 should be incorporated to the main text, not as supplementary.

Figure 1 must be better explained in a new legend. Authors must clarify the difference between chill unit (accumulated chill?) and chilling temperature (temperature of controlled chamber?).

Legend of Figure 9 must be also clarified.

Figure 9 and 10 should be incorporated as supplementary material.

Conclusion section should be completed indicating main implications of the optained results for breeding and production.

6. PLOS authors have the option to publish the peer review history of their article (what does this mean?). If published, this will include your full peer review and any attached files.

Reviewer #1: No

Reviewer #2: No

Reviewer #3: No

---

## [Author Response · Author response to Decision Letter 0]

24 Mar 2020

Dear editor and reviewer,

Please allow us to show our sincere appreciation to your great efforts and kind help for potential publication of our manuscript.

We have carefully studied the editor and reviewers’ comments and discussed with all the co-authors. All the comments are highly valuable, and brought a vast improvement in our original manuscript. Accordingly, the manuscript has been revised thoroughly marked with track change in original draft. Complete answers are provided to the proposed opinions and comments, as follows here. In addition, the language has been fixed by a language editing company named American Journal Editing and the editorial certificate has been uploaded.

Please catch us back anytime if there are any comments not answered properly, looking forward to your hopeful reply. Your time and attention are very much appreciated.

Best regards,

Yue Wang (yuesupermax@163.com)

China Agricultural University

Response to editors

1.To enhance the reproducibility of your results, we recommend that if applicable you deposit your laboratory protocols in protocols.io, where a protocol can be assigned its own identifier (DOI) such that it can be cited independently in the future

I’ve deposited the laboratory protocols in protocols.io and included the DOI link in the Methods section of manuscript.

2.Please ensure that your manuscript meets PLOS ONE's style requirements, including those for file naming

The format has been modified according to PLOS ONE’s style requirements. However, I'm not sure what the line spacing is between the heading and the text. Please catch us back anytime if there are any revision not modified properly.

3.Tighten the abstract and the end of the Introduction to make the aim of the study very clear (abstract and intro) and simplify/shorten the abstract to focus on the key result.

Thanks for the advice, the abstract has been tightened to 216 words, leaving only the key results, and the end of Introduction line 72-76 in previous manuscript is modified to line 90-93 in “Manuscript”

4.Describe the plant material and the experimental design in greater detail using comments from Reviewers 1 and 2.

Thanks for the suggestion. The details about that have been added to line 71-89 and line 96-136. 

5.Revise the reference to adhere to the PLoS format.

It has been revised according to PLOS format.

Response to reviewers

Reviewer#1:

1.Most figures are not clear, if possible, please indicate more clear photos.

Dear reviewer, thanks for the advice. This is the editor’s comments to me “Please disregard comments on figure quality as the compressed version of the paper provided to the reviewer was bad but the figures you submitted are good enough.”

2.The dormant situation in the place of origin and environmental conditions should be provided in order to understand the main factors leading to dormancy.

The authors deeply appreciate the valuable suggestion about our manuscript. I’ve added more details to line 71-89 and line 97-106 in new “Manuscript”.

The three types of dormancy has been identified in our previous research (The regulatory mechanism of chilling-induced dormancy transition from endo-dormancy to non-dormancy in Polygonatum kingianum Coll. et Hemsl rhizome bud. Plant molecular biology. 2019;99(3):205-17.). 

 Our research object is the rhizome bud generated from the seed, and the material collected at the origin is the seed. The rhizome bud was obtained through cultivating the seeds about 50 days under laboratory conditions, and at this time, the rhizome bud entered into endodormancy. The endodormancy can be released by low temperature and then turn into ecodormancy and nondormancy. 

3.Line157-158, the best treatments should be only one.

Thanks for the advice. It has been revised in line 209.

4.Line194-195, A, B, C and D should be shown in Fig 4.

Thanks for the advice. In fact, it is a wrong legend for Fig 4. Now I’ve modified that in line 254-257. and in addition, “Fig 4” has been changed to “Fig 5”.

5.Line201, “are” should be deleted.

Thanks! It has been revised in line 264.

6.Line204, “Clustering dendrogram” should be consistent with that in Fig 5, the Y axis Height should give an explanation in figure legend.

Thanks for the advice. It has been revised in Fig 6, and new legend has been added in Fig 6.

7. Line231, the title does not match what follows, it should be revised.

Thanks for the good suggestion. The title has been modified to “Identification of real hub genes and TFs involved in dormancy transition” in line 297-298.

8. Line252-253, “The results indicated that DAG2 may bind to promoters of the hub genes to regulate bud germination in P. kingianum.” is just a speculation and lack of experimental evidence. Therefore, qRT-PCR analysis of DAG2 expressed in rhizome buds of P. kingianum during chilling and germination processes should be supplemented.

Thanks for the good advice. qRT-PCR analysis of DAG2 for endo-, eco-, nondormant and differentially chilled (15d, 30d, 45d, 60d, 75d, 105d) rhizome buds are supplemented in Fig 10. The description for the result is in line 414-417.

9.Line427, reference 25 (Peter LH, S. Tutorial for the WGCNA package for R: UC Los Ageles; 2014 [) is not cited correctly.

Thanks for the advice. It has been revised to reference 27 according to PLOS format in line 514-516.

10. In Supplementary Fig. 3, “VS” should be lower case letters.

Dear reviewer, thanks for the advice. Do you mean “VS” in the S3_Fig or in the text? There is no “VS” in the figure but I modify that in the text.

11.In addition, the reference format should meet the requirements of PLoS One.

The reference have been modified according to PLoS One format.

Reviewer #2

1.Grammar needs to be thoroughly checked for example line 142.

Thanks for the advice. The language has been polished by AJE. 

2.Line 75 does term mean GO term? 

Yes, term means GO term. I supplement the database in line 163-164.

3. Under methods section, more details about growth conditions such as humidity, watering, light intensity, duration, and weather seeds were grown in soilrite and what media was provided?

The authors deeply appreciate the valuable suggestion about our manuscript. The detail has been supplemented in line 96-136.

4. Line 99: Was day/night temperature same?

Dear reviewer, do you mean the chilling temperature in the “Verification of the chilling models” part? Day/night temperature is the same. The rhizome buds were placed in a controlled environment with a temperature increase from 0 to 10°C at the increment of 1°C every 24 hours, and the increased temperature was maintained for 88 days. Every 11 days, one hundred and twenty rhizome buds were moved to 25°C for sprouting. This part is in line 128-136.

5. How authors classified dormancy as endo or eco or no dormancy. Provide some details.

The authors deeply appreciate the valuable suggestion about our manuscript. This part has been supplemented in line 75-86.

6. Line 132: How transcription start site was determined?

Thanks for the good question, the detail has been supplemented in line 168-172.

7. Line 136: What tissue was used for qPCR analysis?

The bud on the rhizome was used for qPCR analysis. It has been supplemented in line 178-179.

8. Line 139 Instead use qRT-PCR or qPCR everywhere in MS

Thanks for the suggestion. It has been revised in the new manuscript.

9. Format supplementary tables and label title clearly.

Thanks for the good suggestion, I add new legend for S1-S5 Tables (line 581-601) and the label of S4_Table was also supplemented.

Review#3

1.Abstract should be reduced only indicating the key information.

Thanks for the advice, the Abstract has been tightened to 216 words, leaving only the key results.

2.In the Introduction, first paragraph (lines 27-47) should be separated in at least two paragraphs to improve understanding.

Thanks for the advice. This part has been separated in two paragraphs in line 32-52.

3.Objectives of the work should be better clarified eliminating the references (page 4 line 96).

Dear reviewer, Thanks for the suggestion. Do you mean the paragraph named “Relationship between chilling accumulation and sprouting percentage”? I’ve added details in this part in line 121-126.

4.Description of methodology should be improved. The Plant material used must be clarified indicating if it is possible any reference. Description of RT-PCR must be completed. It is very important the description of the assayed genes.

The authors deeply appreciate the valuable suggestion about our manuscript.. The details about plant materials have been supplemented in line 96-136. Details of RT-PCR has also been supplemented in line 177-191, and the description for the result is in line 414-417.

5.Figure S2 should be incorporated to the main text, not as supplementary.

Thanks for the advice. It has been modified to Fig 1 in the new manuscript.

6.Figure 1 must be better explained in a new legend. Authors must clarify the difference between chill unit (accumulated chill?) and chilling temperature (temperature of controlled chamber?).

Thanks for the good question. Chill unit describes the efficiency for the cold temperature on releasing the bud dormancy and it was calculated on the basis of the average slope of all the chilling-response curves (Fig1 in the new manuscript) under each chilling temperature. 

Chilling temperature used in this study is -2, 0, 2, 4, 6, 8, 10, 12 and 14°C. Through the experiment “Chill unit confirmation”, we obtained the corresponding relation between “chill unit” and “cold temperature” (Fig2 in the new manuscript). 

We define the chilling accumulation for P. kingianum rhizome buds within an hour at a certain temperature as being equal to the chill unit for the temperature obtained above, thus, we can figure out the chill accumulation over a period of time.

I added the details in line 108-109, 122-123 and 234-239.

7.Legend of Figure 9 must be also clarified.

Thanks for the advice. The new legend has been supplemented for S6 Fig (the new manuscript) in line 574-577.

8.Figure 9 and 10 should be incorporated as supplementary material.

Thanks for the suggestion. Figure 9 and Figure 10 have been modified to S6 Fig and S7 Fig respectively.

9.Conclusion section should be completed indicating main implications of the optained results for breeding and production.

Thanks for the good advice. The sentence “the modules indicated that chilling accumulations of 1940-964 Ca are required from the 37th to the 50th posttreatment warming days to achieve 90% sprouting, which can help guide the breeding of P.kingianum.“ has been supplemented in line 435-437.

---

## [Editor Report · Decision Letter 1]

3 Apr 2020

Characterizing rhizome bud dormancy in Polygonatum kingianum : development of a novel chill models and determination of dormancy release mechanisms by weighted correlation network analysis

PONE-D-20-00745R1

Dear Dr. Wang,

We are pleased to inform you that your manuscript has been judged scientifically suitable for publication and will be formally accepted for publication once it complies with all outstanding technical requirements.

The revision now largely meets the criteria for publication in PLoS ONE. Congratulations.

However, I did notice some room for improvement as the manuscript still contains a number of typos and approximation. For example the protocol here https://www.protocols.io/view/a-method-for-motif-based-sequence-alignment-analys-bdwki7cw is full of typos and the link to meme-suite (note the hyphen) is broken. I would really appreciate you to take the time to go through the manuscript at the proofing stage very carefully and make sure you are doing justice to your hard work.

With kind regards,

Alexandre Fournier-Level, Ph.D.

Academic Editor

PLOS ONE

---

## [Editor Report · Acceptance letter]

9 Apr 2020

PONE-D-20-00745R1 

Characterizing rhizome bud dormancy in Polygonatum kingianum : development of novel chill models and determination of dormancy release mechanisms by weighted correlation network analysis 

Dear Dr. Wang:

I am pleased to inform you that your manuscript has been deemed suitable for publication in PLOS ONE. Congratulations! Your manuscript is now with our production department. 

With kind regards,

on behalf of

Dr. Alexandre Fournier-Level 

Academic Editor

PLOS ONE